Trueness and precision of complete denture digital impression compared to conventional impression: an in vitro study

http://orcid.org/0000-0002-8180-8903 Al-Dulaijan Yousif A. 1 yaaldulaijan@iau.edu.sa
Alalawi Haidar 1
Gad Mohammed M. 1
Al-Qarni Faisal D. 1
Fouda Shaimaa M. 1
Ellakany Passent 2
1 Department of Substitutive Dental Sciences, College of Dentistry, Imam Abdulrahman Bin Faisal University , Dammam , Saudi Arabia
2 Division of Prosthodontics, School of Dentistry, University of Alabama-Birmingham , Birmingham, Alabama , United States
Abu Hasna Amjad
Electronic publication date: 2025 Feb 26
Publication date: 2025
Volume: 13
Electronic Location ID: e19075
Received 2024 Nov 1; Accepted 2025 Feb 10
Copyright: © 2025 Al-Dulaijan et al.
Copyright year: 2025
Copyright holder: Al-Dulaijan et al.
License: This is an open access article distributed under the terms of the Creative Commons Attribution License, which permits unrestricted use, distribution, reproduction and adaptation in any medium and for any purpose provided that it is properly attributed. For attribution, the original author(s), title, publication source (PeerJ) and either DOI or URL of the article must be cited.
License URL: https://creativecommons.org/licenses/by/4.0/

Keywords: CAD-CAM, Complete denture, Impression, Trueness, Precision

Funding: The authors received no funding for this work.

==============================
Background

This study aimed to compare the precision and trueness of digital impressions of the edentulous arch made with different scanners to conventional physical impressions.

Methods

A total of 40 impressions of a completely edentulous maxillary arch model (n = 10) were made using different digital impressions with an extraoral scanner, E3 3Shape desktop scanner, as the reference scan, intraoral scanner (TRIOS IOS, and Medit IOS) and Vinyl Polysiloxane impressions (VPS) impression using a Computer-Aided Design and Computer-Aided Manufacturing (CAD-CAM) custom tray. The VPS impression was scanned with the desktop scanner to produce standard tessellation language (STL) files for comparison with the digital impressions made by the Desktop and intraoral scanners. The STL files were super-imposed to a desktop scan and to each other with the same group using Geomagic Control X Software to assess the trueness and precision, respectively. A t-test was conducted for statistical analysis with a significance level of 0.05.

Results

The overall trueness, Medit had the highest deviation compared to the VPS and TRIOS groups with a P value of 0.0013 and <0.0001, respectively. In terms of overall precision, TRIOS had a lower deviation than the VPS group, with a P value of 0.0002. The TRIOS and Medit groups had statistically comparable results. The desktop scanner showed the highest precision in digitizing completely edentulous cases, followed by the TRIOS scanner. The Medit scanner’s trueness had the highest deviation compared to the VPS and TRIOS groups.

Introduction

With the rise of Computer-Aided Design and Computer-Aided Manufacturing (CAD-CAM), digital impressions have emerged as a clinically valid alternative to traditional impressions. The benefits include the capability to assess and alter the scanned area, a decrease in the use of impression materials, and reduced costs for model storage. In addition, it is considered a better and faster way of communication with the dental laboratory (Li et al., 2022). It can be used primarily in fabricating crowns, fixed partial dentures, implant-supported crowns, and removable partial and complete prostheses (Ahlholm et al., 2018; Ellakany et al., 2024).

Conventionally, a master model is required to fabricate any fixed or definitive prosthesis. In the case of complete denture construction, several impression techniques (one- or two-step) and materials were used to register the functional border and intaglio surface of the denture base (Jayaraman et al., 2018).

Digitally, a complete denture (CD) could be fabricated by scanning a conventional impression, a cast, or the edentulous arch intraorally, followed by making a maxillomandibular record, designing the denture base, and arranging the artificial teeth virtually using a CAD program (Kattadiyil, Goodacre & Baba, 2013; Goodacre & Goodacre, 2018; Chebib et al., 2019; Wang et al., 2021). Studies have reported that making impressions using intraoral scanners is more comfortable for patients. Hence, they reduce anxiety and nausea compared to conventional impressions (Kattadiyil et al., 2015).

Accuracy is defined by trueness and precision, where trueness is the deviation from the actual dimensions of the scanned object. In contrast, precision is the consistency of the repeated scans (International Organization for Standardization, 1994; Ender & Mehl, 2013). Therefore, the scanner is required to possess high trueness and precision.

Precision and trueness are particularly critical for edentulous arches due to their direct impact on complete dentures’ fit, stability, and retention. Unlike dentate arches, where the presence of teeth provides defined landmarks for impression-making, edentulous arches rely heavily on accurate replication of soft tissue contours and sulcus depth. Any deviation in these parameters can result in ill-fitting dentures, leading to compromised retention, discomfort, and functional limitations for the patient (Kattadiyil et al., 2015; Al Hamad & Al-Kaff, 2023). Precision ensures consistent reproduction of the arch’s morphology across multiple impressions, which is vital for achieving repeatable results in denture fabrication. On the other hand, trueness ensures that the impressions closely replicate the actual anatomy of the edentulous arch, reducing the need for chairside adjustments and enhancing clinical efficiency (Ender & Mehl, 2013; Hack & Patzelt, 2015).

For the dentate model, several in vitro studies were reported to evaluate the accuracy of the intraoral scanners (Nedelcu & Persson, 2014; Hack & Patzelt, 2015; Su & Sun, 2015; Jeong et al., 2016; Renne et al., 2017). Accuracy in terms of precision and trueness varied between scanners; however, it was clinically accepted (International Organization for Standardization, 1994; Ender & Mehl, 2013; Hack & Patzelt, 2015). Nedelcu & Persson (2014) suggested using digital scanning carefully and with limitations to short-span dental prostheses.

Previous studies reported that the accuracy of scanning edentulous tissues relies on various factors such as flexibility, mobility, and dimension of soft tissues (Ender & Mehl, 2013; Kattadiyil et al., 2015; Chebib et al., 2019; Lo Russo et al., 2020; Hack et al., 2020; Al Hamad & Al-Kaff, 2023).

Comparing conventional to digital impressions is significant, reflecting the ongoing shift toward digital workflows in prosthodontics. This comparison provides critical insights into the strengths and limitations of each approach, guiding clinicians in selecting the most appropriate method for individual cases. Chebib et al. (2019) compared the trueness of the intraoral scanner to several conventional impression materials, irreversible hydrocolloid, polyvinyl siloxane (PVS), PVS modified with zinc oxide eugenol (ZOE), and ZOE for making maxillary edentulous jaw impression. They reported similar deviations as PVS and PVSM. Another study by Li et al. (2022) compared the accuracy of two intraoral scanners with four different conventional impression materials and reported comparable results. Using specially designed software, Kalberer et al. (2021) assessed the vertical and horizontal deviations. The PVS impression and the digital scan did not significantly differ and did not show significant error differences. However, PVS was more accurate than the digital scan in the anterior region. At the inner seal and approximately 2 mm from the border molded impression, the PVS proved to be substantially more accurate than the digital scan. This finding also agreed with the study by D’Arienzo, D’Arienzo & Borracchini (2018), which supported the compressive effect of conventional impressions over the static one of intraoral scanning. Previous studies stated that the lips and cheeks need to be significantly retracted throughout the scanning process, which could be the causative factor in reducing the accuracy of the peripheral seal (Chebib et al., 2019; Hack et al., 2020). A digitally fabricated complete denture has been shown to exhibit robust retention, even in the absence of a functional impression derived from direct scanning of the jaws (Lo Russo & Salamini, 2018). Therefore, the sealing effect from custom-formed denture borders might not be as significant as the surface tension between the denture base and the underlying tissue (Bosniac, Rehmann & Wöstmann, 2019).

These conflicting findings need to be further studied to provide valid proof of using digital impressions for edentulous arches, considering all aforementioned variables and determining the optimum method of recording the extent of the sulcus and peripheral seal to get sufficient retention. Retention is the core of the success and longevity of complete dentures inside the patient’s mouth. Thus, the present study aimed to evaluate the trueness and precision of different impressions made for a completely edentulous model using digital and conventional approaches. The study’s null hypothesis stated that the digital impressions scanned by desktop and intraoral scanners exhibit similar trueness and precision compared to conventional VPS impressions.

Materials and methods

Model preparation

An edentulous maxillary model (original model) was used to conduct the present study. The original model was scanned using an E3 3Shape desktop scanner (3Shape, Inc., Copenhagen, Denmark) to create a standard tessellation language (STL) file. The scanned model was modified to include three spheres (Fig. 1) using Fusion 360 software (Autodesk, San Francisco, CA, USA). Two spherical objects were positioned at the crest of the ridge above each tuberosity and another one on the anterior crest of the ridge near the midline. These objects were employed to place a measuring guide accurately following various impression techniques on the model to guarantee that measurements were taken at precise positions (Goodacre et al., 2016).

Figure 1 Study flowchart.

The model was printed with a model resin material (NextDent Model 2.0; NextDent B.V., Soesterberg, The Netherlands). The printed (reference) model was used to conduct the experiment, which included four groups according to the impression techniques. To reduce variability, all the scans were performed by the same experienced operator (H.A.) (Ozkurt, Dikbas & Kazazoglu, 2013).

Sample size and study groups

A sample size calculation revealed the need for ten scans per impression following a previous study (Goodacre et al., 2016). The study included four study groups (n = 10) of digital impressions using the extraoral desktop scanner of the edentulous model as the control group (Desktop), two intraoral scanners (TRIOS and Medit), and desktop scans of vinyl polysiloxane conventional impression (VPS). In total, 40 scans were performed and analyzed across the four groups (Fig. 1).

Digital impressions

The extraoral desktop scanner (3Shape E3 scanner; 3Shape, Copenhagen, Denmark) was used to scan the reference model, obtaining a reference scan to assess the trueness of the other methods of complete denture impression.

Then, the reference model was scanned again ten times to produce STL files of the Desktop group. Similarly, two intraoral scanners, 3Shape TRIOS 3 (3Shape, Copenhagen, Denmark) and Medit i700 (MEDIT Corp., Seoul, Korea), were used to produce ten scans of the reference model. The scanning sequence was standardized for both scanners. From the left side of the arch, the U-shaped portion of the alveolar ridge was first scanned. The alveolar ridge’s buccal or labial aspect was then scanned, followed by the palatal region. Finally, the palate was thoroughly scanned by moving zigzag-style from the incisive papilla to the soft palatal region (Li et al., 2022).

Scans of VPS conventional impression

The reference model was scanned with the extraoral desktop scanner to design a digital custom tray using the 3Shape Dental System (3Shape, Copenhagen, Denmark). The tray material was 2 mm thick, and the impression material had a spacer of 2 mm. The tray was printed with a tray resin material (NextDent Tray Resin, NextDent B.V., Soesterberg, The Netherlands).

Ten Vinyl Polysiloxane impressions (VPS) (3M™ Imprint™ 3 VPS Impression Material-Regular Body; 3M, Saint Paul, MN, United States) were made using a dispensing gun with a printed customized impression tray (Fig. 1) to make the conventional impression. After ten impressions were made, the VPS impressions were scanned using the desktop scanner (3Shape, Inc., Copenhagen, Denmark) using the same method described above for model scanning, and then the STL files were generated. Following that, each scan of the conventional impression was inverted and trimmed using Autodesk Meshmixer Software (Autodesk Inc., San Rafael, CA, United States) to produce a model comparable to the digital scans of the reference model.

Measurement of trueness and precision

The digital impression using a desktop scanner (Desktop) was used as a reference scan for trueness measurements. All other groups’ STL files were superimposed to the reference scan in the metrology software program (Geomagic Control X, version 2018.0.1; Geomagic Inc., 3D Systems, Rock Hill, SC, USA). This software uses an initial alignment feature to superimpose each sample to the reference scan. Then, best-fit alignment is utilized to produce the most accurate superimposition. After that, 3D comparisons were performed on four different areas (residual ridge, palate, vestibule, and overall) (Fig. 2). Root mean square (RMS) value was used to compare the deviation of the resultant scans to the reference scan, similar to previously published studies (Li et al., 2022; Abualsaud & Alalawi, 2022).

Figure 2 3D comparisons of trueness measurements.

3D comparisons of trueness measurements of the four areas (residual ridge, palate, vestibule, and overall) after the STL files were superimposed to the reference scan.

For precision evaluations, the STL files within each experimental group were superimposed on each other using the metrology software program (Geomagic Control X, version 2018.0.1, Geomagic Inc., 3D Systems, Rock Hill, SC, USA). This software uses an initial alignment feature to superimpose each sample to the reference scan. Then, best-fit alignment is utilized to produce the most accurate superimposition. After that, 3D comparisons were performed on four different areas (residual ridge, palate, vestibule, and overall) (Fig. 3). Then, the square root of the mean of the squares (root mean square [RMS]), which is a metric provided by the software as a representative measure of the deviations between the reference scan and the tested superimposed scan, was used to reflect the average magnitude of differences, incorporating both positive and negative variations to provide a single, comprehensive value (Li et al., 2022; Abualsaud & Alalawi, 2022).

Figure 3 3D comparisons of precision measurements.

3D comparisons of precision measurements of the four areas (residual ridge, palate, vestibule, and overall) after the STL files were superimposed to the reference scan.

The trueness and precision of different impression technique results were expressed as means and standard deviations (SD). Data were analyzed with a repeated-measures analysis of variance followed by a comparison with a student t-test using a statistical software program (JMP®, Version 16; SAS Institute Inc., Cary, NC, USA).

Results

The mean values and standard deviations (SDs) for trueness and precision across the tested groups are summarized in Tables 1 and 2. For trueness, the Medit group consistently showed the highest RMS values across all anatomical areas. At the residual ridge, Medit exhibited the largest RMS value, followed by the VPS and TRIOS groups, with statistically significant differences observed (P = 0.0097 and P = 0.0002, respectively). Similarly, at the palatal area, Medit had the highest RMS value, followed by TRIOS and VPS, with significant differences (P = 0.0126 and P = 0.0002, respectively). At the vestibular area, TRIOS demonstrated the lowest RMS value, followed by VPS and Medit, with a highly significant difference (P < 0.0001). Regarding overall trueness, Medit displayed the highest RMS values across all areas, followed by VPS and TRIOS (P < 0.0001 and P = 0.0013, respectively). These findings highlight that the Medit group exhibited the highest RMS values in all scanned areas.

Table 1 The trueness (µm) of the various complete denture impression techniques compared to desktop.

Impression technique	Ridge
mean ± SD	Palate
mean ± SD	Vestibule
mean ± SD	Overall
mean ± SD	
VPS	38.191 ± 12.580 (A)	34.764 ± 13.738 (A)	49.636 ± 8.209 (B)	39.509 ± 11.029 (A)	
Medit	49.413 ± 2.667 (B)	53.660 ± 32.738 (B)	52.800 ± 2.393 (B)	52.138 ± 2.272 (B)	
TRIOS	31.940 ± 5.889 (A)	41.520 ± 6.950 (A)	34.940 ± 3.552 (A)	34.870 ± 4.946 (A)	
P value	0.0010	0.0010	<0.0001	0.0002	
CI	34.99–43.27	37.73–46.87	41.81–49.08	37.55–45.24	
Note:

Levels not connected vertically by the same letter are significantly different.

Table 2 The precision (µm) of the various complete denture impression techniques.

Impression technique	3D Ridge
mean ± SD	3D Palate
mean ± SD	3D Vestibule mean ± SD	3D Overall
mean ± SD	
VPS	39.050 ± 10.683 (A)	36.230 ± 12.988 (A)	51.190 ± 4.282 (A)	40.930 ± 9.451 (A)	
Medit	34.329 ± 7.955 (A,B)	37.900 ± 13.883 (A)	34.557 ± 7.506 (B)	35.414 ± 7.127 (A,B)	
TRIOS	28.889 ± 3.458 (B)	26.356 ± 4.034 (B)	32.678 ± 5.858 (B)	29.567 ± 3.386 (B)	
Desktop	8.617 ± 0.906 (C)	7.533 ± 0.398 (C)	9.725 ± 0.612 (C)	8.658 ± 0.643 (C)	
P value	<0.0001	<0.0001	<0.0001	<0.0001	
CI	21.51–30.81	20.04–30.24	25.14–36.16	22.28–31.78	
Note:

Levels not connected vertically by the same letter are significantly different.

Precision analysis revealed that the desktop scanner achieved the lowest RMS values across all anatomical areas, indicating superior performance. At the residual ridge, Trios demonstrated a significantly lower RMS value than VPS (P = 0.0022), while its results were statistically comparable to Medit. At the palatal area, TRIOS exhibited lower RMS value than Medit and VPS (P = 0.0165 and P = 0.0239, respectively), with the latter two groups showing no significant difference. At the vestibular area, VPS recorded the highest RMS value (P < 0.0001), whereas TRIOS and Medit had comparable results. For overall precision, the Desktop scanner showed the lowest deviation, followed by TRIOS, while VPS had the highest (P < 0.001). TRIOS and Medit results were statistically comparable, while VPS and Medit had the highest RMS values in specific areas.

In summary, the Medit group exhibited the highest RMS values in trueness across all anatomical areas, suggesting lower accuracy. Conversely, the desktop scanner outperformed other groups in precision, with the lowest RMS values observed in all regions. These results underscore the variability in performance among the tested impression techniques and highlight the importance of selecting appropriate methods based on clinical requirements.

Discussion

This study aimed to compare the accuracy of different digital impressions with conventional ones regarding trueness and precision. Based on the results, the null hypothesis was partially rejected. Also, it was observed that a lower reported precision value reflects higher accuracy, as precision is defined by the consistency of repeated measurements and quantified by the degree of variation or deviation. This distinction is critical for interpreting the study’s findings, as lower deviations signify greater reproducibility and accuracy of the scanning system.

TRIOS exhibited the lowest RMS values for trueness, comparable to VPS, particularly in the whole arch, ridge, and vestibular areas. Conversely, the Medit scanner demonstrated the highest RMS values, highlighting a lower trueness value than other groups. These findings align with prior research (Michelinakis et al., 2020), which identified similar patterns of accuracy differences between intraoral scanners.

Desktop scanners showed the lowest RMS values in precision, indicating superior reproducibility across all regions. This high level of precision can be attributed to the structured light or laser beam technology employed in desktop scanners, which provides accurate data capture over larger spans compared to intraoral scanners (Su & Sun, 2015; Mangano et al., 2019). Meanwhile, VPS exhibited the highest precision RMS values, likely due to the compressive nature of the impression material and the manual application pressure, as suggested in earlier studies (Masri et al., 2020).

These findings underscore the importance of interpreting accuracy metrics correctly: lower precision RMS values indicate higher reproducibility, while higher trueness RMS values denote less accurate correspondence to the actual anatomy. This distinction is essential for practitioners evaluating the suitability of various impression techniques for clinical use.

Digital impressions must achieve at least the same quality and accuracy as conventional procedures to replace the traditional impression and stone cast (Abualsaud & Alalawi, 2022). Numerous methods have been utilized to assess the precision of IOS. Nevertheless, employing reference scan information from a high-precision industrial scanner is still considered the most reliable approach for evaluating accuracy (Christensen, 2009; O’Toole et al., 2019).

The selection of the 3Shape E3 desktop scanner, the 3Shape TRIOS 3 intraoral scanner, and the Medit i700 intraoral scanner for this study were based on their demonstrated performance and extensive evaluation in prior research. These scanners were chosen because they represent advanced digital technologies widely validated for their accuracy in scanning dental models, particularly regarding trueness and precision (Ahlholm et al., 2018; Hack et al., 2020). Also, vinyl polysiloxane impressions (VPS) impression material was selected as the conventional benchmark due to its long-standing reputation as a gold standard in prosthodontics for its dimensional stability and fine detail reproduction (D’Arienzo, D’Arienzo & Borracchini, 2018; Al Hamad & Al-Kaff, 2023). PVS’s reliability allows for a robust comparison between traditional and digital impression methods.

A single experienced operator made all scans and impressions in this study to minimize variability caused by differences in operator skill. Operator variability is a significant factor affecting the trueness and precision of digital impressions, as prior studies have demonstrated differences in scan accuracy based on operator technique, experience, and familiarity with the equipment. By standardizing the scanning process (scanner, lighting, temperature, and humidity) with a single operator, the study ensures that any observed differences in the results are attributable to the scanning systems or methods rather than inconsistencies in operator performance.

Based on the trueness finding of the present study, TRIOS exhibited the lowest RMS values (among the whole arch, ridge, and vestibular areas), which is comparable to VPS. At the same time, the Medit impression showed the highest RMS value among the tested groups. This finding, in disagreement with previous studies, reported higher accuracy of VPS impressions compared to digital impressions made by intraoral scanners (Ender & Mehl, 2013; Malik et al., 2018; Seo & Kim, 2021). On the other side, Michelinakis et al. (2020) found that the trueness of TRIOS and Medit intraoral scanners are comparable in recording the dentate casts. This deviation between findings in the literature might result from several factors that affect the accuracy of digital impressions, such as the operator’s proficiency, the scanning technique, the scanning procedures, the location of the scanning camera, and the number of captured images during scanning (Hayama et al., 2018). Additionally, the highest RMS values of the trueness of VPS might result from the nature of conventional impressions, where both the pressure generated on the tissues or the cast during impression application and the difference in pressure generated manually by the operator can lead to this deviation (Masri et al., 2020).

Desktop scanners presented the highest precision readings among all tested groups in the current study. This comes in agreement with several previous studies (Su & Sun, 2015; Ellakany, Aly & Al-Harbi, 2023). The higher precision might be due to the higher accuracy of desktop scanners in recording digital impressions of larger dimensions than intraoral scanners (Mangano et al., 2019). Using a laser beam or structured light in desktop scanners favors this high accuracy, resulting in a minimal error rate, unlike the case of the limited optical field of intraoral scanners. Moreover, other factors related to intraoral oral impressions, such as the presence of saliva, tongue, the wetness of the lens, and the scanned surface reflections, can reduce the accuracy of intraoral impressions. In addition, desktop scanners are capable of capturing multiple numbers of images automatically, unlike the stitching method of intraoral scanners, which might result in a pronounced deviation, particularly in scanning a full arch or long-span regions (Abduo, 2019; Ellakany, Aly & Al-Harbi, 2023). TRIOS demonstrated significantly different trueness values compared to the Medit scanner, contrasting with findings reported in another study (Michelinakis et al., 2020). Zarone et al. (2020a) reported different findings where the TRIOS intraoral scanner showed better accuracy than a desktop scanner (DScan 3). This might be attributed to using different impression materials, polysulphide, or desktop scanners tested. Meanwhile, the VPS impression exhibited the lowest precision in the 3D overall comparison area among all tested groups in the current study. This might be justified by the same reasons of the compressive nature and pressure exerted during the impression (Hayama et al., 2018; Masri et al., 2020).

However, the vestibular area showed a high deviation range in trueness and precision. This might be due to the difficulty in capturing the sulcular depth sulcus and the borders, especially using intraoral scanners. In the intraoral scanning, the lips and the oral tissues should be stretched and retracted to insert the scanning head, which would affect the accuracy of the impression periphery and extension (Masri et al., 2020). D’Arienzo, D’Arienzo & Borracchini (2018) compared the clinical accuracy of TRIOS intraoral scanners to a reference edentulous maxillary model fabricated from conventional alginate impression and scanned by desktop scanner (3Shape D1000). The greatest discrepancy was reported in peripheral regions, such as dental vestibules and soft palates, which aligns with the current results. This may be due to the high compression forces applied by the alginate impression material on these tissues, unlike the intraoral scanners. Consequently, it has been noted that intraoral scanners can be used for recording preliminary impressions of edentulous arches but not for definitive impressions that require selective pressure to obtain retentive prostheses. Similar findings were reported by Zarone et al. (2020b), in which the precision and trueness of different scanning patterns were assessed using a TRIOS 3Shape intraoral scanner. The highest distortion was reported in the palatal rugae regions of the edentulous maxillary arch.

Desktop scanners are a great addition to digital prosthetic dentistry, capturing accurate impressions for completely edentulous ridges and simulating the contour of the oral mucosa. As an alternative to conventional impressions, TRIOS scanners can be used as a valid intraoral digital impression technique with an acceptable tissue tolerance range without denture sinking.

The clinical thresholds for the deviation in denture impression accuracy have been discussed in prior studies, typically ranging between 50–200 microns for complete dentures, depending on the area of the impression and the clinical application (Ender & Mehl, 2013; Hack & Patzelt, 2015). For instance, deviations below 100 microns in critical areas such as the intaglio surface or borders are generally considered clinically acceptable, as they maintain adequate retention, stability, and patient comfort. While both methods yielded results within clinically acceptable limits, the slight discrepancies in accuracy between techniques suggest that practitioners should consider specific patient factors (e.g., tissue mobility, ridge resorption) when choosing an impression method to optimize outcomes. For instance, areas with higher deviations in digital impressions (e.g., sulcus depth) may necessitate selective adjustments during the denture fabrication process to ensure optimal fit.

The present study’s findings should be presented with cautions related to in vitro study limitations until these findings are proven with further in vivo studies. This in vitro study has a few limitations, including the borders being arbitrarily defined in the conventional impression group as the deepest part of the vestibular area without using the border-molding processes. Additionally, the protocol needs validation in clinical settings due to the possibility of the patient mobility, salivation, and border-molding technique that could alter the precision and trueness of the completely edentulous impression-making techniques. Therefore, further clinical studies are needed to verify the present results.

Conclusions

The following conclusions were drawn within the limitations of this in vitro study. First, Extraoral desktop scanners demonstrated the highest precision in digitizing completely edentulous cases, establishing their reliability as a gold standard for digital impressions in prosthodontics. Second, the TRIOS intraoral scanner and VPS impressions exhibited comparable trueness, suggesting that digital impressions using the TRIOS scanner can serve as a clinically viable alternative to conventional methods for fabricating complete dentures. Third, the Medit intraoral scanner, while providing adequate results, showed the highest deviations in trueness. These findings highlight the need to carefully consider scanner choice based on specific clinical requirements and patient needs.

This study provides novel insights into the comparative accuracy of different impression techniques for completely edentulous cases. The findings support the growing adoption of digital workflows in prosthodontics by demonstrating the clinical equivalence of TRIOS scanners to conventional VPS impressions in trueness. Furthermore, identifying precision advantages with desktop scanners reinforces their role in complex cases requiring high fidelity.

These results underscore the potential of digital impression systems to enhance efficiency, patient comfort, and treatment outcomes in prosthodontic practice while paving the way for further studies to refine digital workflows for edentulous arches.

Supplemental Information

Supplemental Information 1 Trueness and precision of CD Impression raw data.

Additional Information and Declarations

Competing Interests

The authors declare that they have no competing interests.

Author Contributions

Yousif A. Al-Dulaijan conceived and designed the experiments, performed the experiments, analyzed the data, prepared figures and/or tables, authored or reviewed drafts of the article, and approved the final draft.

Haidar Alalawi performed the experiments, prepared figures and/or tables, and approved the final draft.

Mohammed M. Gad conceived and designed the experiments, performed the experiments, authored or reviewed drafts of the article, and approved the final draft.

Faisal D. Al-Qarni performed the experiments, authored or reviewed drafts of the article, and approved the final draft.

Shaimaa M. Fouda conceived and designed the experiments, analyzed the data, prepared figures and/or tables, and approved the final draft.

Passent Ellakany conceived and designed the experiments, analyzed the data, authored or reviewed drafts of the article, and approved the final draft.

Data Availability

The following information was supplied regarding data availability:

The raw measurements are available in the Supplementary File.

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
