# Peer review of "Trueness and precision of complete denture digital impression compared to conventional impression: an in vitro study"

_PeerJ, doi:10.7717/peerj.19075_

## Round 0.1 · original submission · Major Revisions

Dear authors,

Thank you for submitting your manuscript addressing the precision and trueness of digital versus conventional impressions for edentulous arches. While your study explores a significant topic and employs robust methodology, major revisions are required to enhance its clarity, rigor, and impact. These include refining English grammar and phrasing, updating statistical analyses to ensure validity, and improving data reporting. Additionally, the manuscript would benefit from better organization of the introduction and discussion, along with clearer connections between your findings and their clinical implications. We encourage you to carefully address the detailed comments provided by the reviewers to strengthen your work.

Reviewer 1 ·

Basic reporting

The manuscript is generally well-written in professional English, but minor grammar and phrasing adjustments could improve clarity. The introduction is comprehensive and contextualizes the study within the existing literature, explaining the relevance of digital impressions in prosthodontics, particularly for complete dentures. The background is supported by recent references, making it suitable for understanding the study's purpose and significance.

Experimental design

The authors define a clear research question that addresses a knowledge gap about the precision and trueness of digital versus conventional denture impressions.

The methodology is detailed and appropriate, particularly the use of STL files and software to assess trueness and precision, which is standard practice in such comparative studies. The sample size calculation is clearly explained, aligning with previous studies to ensure statistical reliability.

However, one area for improvement is to specify the rationale for selecting the particular intraoral and desktop scanners and the impression materials used, as additional explanation would clarify why these tools were ideal for the study's aims.

Validity of the findings

The study’s findings are well-supported by statistical analysis, which appears robust and suitable for the data type.

Additional comments

The manuscript is well-structured and meets the key requirements for clarity and relevance. The study contributes valuable insights into the accuracy of digital impressions for denture fabrication and supports the desktop scanner as a reliable tool for precision in edentulous cases.

Reviewer 2 ·

Basic reporting

Dear Authors,
Thank you for the opportunity to review your manuscript. Your research addresses an important topic in our field, and the work presents valuable insights into the evolving landscape of digital and conventional impressions. Overall, the manuscript is well-written, and the English used is professional and clear.

The background provided establishes a solid foundation for understanding the study.

However, there are areas where the manuscript could benefit from improvements. In my opinion, the Discussion section would be strengthened with better organization and integration of findings. Currently, it moves between comparisons and factors without a clear structure, which makes it challenging to follow. Additionally, while the references to other studies are appreciated, their relevance to the current results is not always fully explained. Providing a more direct connection between your findings and those in the literature would enhance the clarity and impact of this section.
Moreover, some claims, such as "Trios scanners can be used as a valid intraoral digital impression technique with an acceptable tissue tolerance range," appear too broad given the study’s in vitro nature. It is essential to emphasize that these findings are based on an in vitro setup and require clinical validation before such recommendations can be generalized.

Experimental design

The study design is appropriate for addressing your research question, but there are a few methodological aspects that could be elaborated on to improve the clarity and robustness of your work:

- Operator Bias: Scans were performed by a single experienced operator (H.A.). While this minimizes variability caused by differences in operator skill, it introduces the possibility of bias. Including multiple operators or discussing how this limitation impacts the generalizability of the findings would strengthen this section.
- Statistical Analysis: The use of repeated-measures ANOVA and t-tests is appropriate. However, the manuscript does not specify how potential confounders, such as scanner calibration or time intervals between scans, were managed. Addressing these potential sources of variability would enhance the reliability of your conclusions. Additionally, the results focus heavily on statistical significance (P-values) without reporting effect sizes or confidence intervals. Providing these metrics would allow readers to assess the clinical relevance of your findings, even when statistical differences are detected.
- Numerical Reporting: While means and standard deviations (SDs) are mentioned in relation to Tables 1 and 2, the text itself lacks key numerical details about deviations in different areas. Including these numbers in the text would improve readability and reduce the need for readers to constantly consult the tables.

Validity of the findings

The manuscript demonstrates technical rigor, but the clinical implications of the findings could be more thoroughly discussed. For example, it is unclear how the observed deviations relate to clinical thresholds of acceptability or how they might impact patient outcomes. A deeper exploration of these points would make the results more relatable and impactful for practitioners.

Additionally, while the study addresses trueness and precision in detail, the conclusions do not sufficiently highlight the novel contributions of the work or its significance to the field. In my opinion, what is unique about your findings and their potential to advance the field needs to be emphasized.

Reviewer 3 ·

Basic reporting

This study evaluates the precision and trueness of digital impressions and conventional impressions for edentulous arches describing a scanning sequence highly standardized and performed under controlled conditions. However, major revisions need to be performed such as English grammar, statistical tests, and consequently, update the data obtained.

Introduction:
-The authors highlight the importance of the trueness and precision of scanners properly, However, the introduction contains excessive background information, making it difficult to discern the specific aims of the study.
-The transition from conventional techniques to digital approaches is overly detailed and not consistently linked to the main objective.
-In addition, I would suggest that authors emphasize why precision and trueness matter for edentulous arches and why comparing conventional to digital impressions is significant
-For the hypothesis, it is feasible if the authors keep only one null hypothesis.
- The grammatical of the manuscript must be revised, and sentences should be simplified to clarify and to avoid correct grammatical errors. E.g.: "A digitally created complete denture reportedly demonstrates strong retention, even without a functional impression from the jaws’ direct scan". This statement is overly verbose and unclear.
-The definition of PVS must not be repeated (lines 81 and 86). The authors should only describe it in the first mention time of the manuscript.
- Some sentences are excessively burdened with citations, which disrupts readability (e.g., Lines 77–79, 81–84).
- Inconsistent citation styles, e.g., "D'Arienzo et al. (D Arienzo, D Arienzo & Borracchini, 2018)" contains duplicate author names and missing apostrophes.

Experimental design

-The authors mentioned an edentulous maxillary model was scanned. However, which criteria do the authors establish for this edentulous maxillary? A model without any anatomic accidents?
-I would suggest mentioning “The study included four study groups (n=10) of digital impressions using the extraoral Desktop scanner of the edentulous model as the control group (Desktop), two intraoral scanners (Trios and Medit), and desktop scans of Vinyl Polysiloxane conventional impression (VPS). In total, 40 scans were performed and analyzed across the four groups (Fig 1).” To clarify this description, remove the “N=40” of “(…) to conduct the present study with total scans N=40”.
-I got confused if the anatomic scan was scanned ten times or if the reference model (stl from the anatomic model) was scanned ten times. To clarify, I suggest referring to the “printed model” as a “reference model”. Moreover, the sentence “Then, the model was scanned again…” should be changed to “Then, the reference model was printed and scanned ten times…” if I understood correctly.
-The data were analyzed to check the normality? If yes, which test did the authors apply for? The repetitive t-test can lead us to commit the type I (α) error of statistics, being indicated to analyze the variance only between two groups.
-In the last paragraph of the Measurement of trueness and precision section, please, correct “Standard of Deviations “ for standard deviations.
-The Figure 2 has a poor quality. Please, improve this image.

Validity of the findings

Results:
-The authors describe the means as “deviations” which may confuse the reader. To improve the data reporting, the authors must replace the deviations for means, since deviations are related to standard deviations (SD).
-The mean should have the same number of digits following the decimal point as the standard deviation has.

-The description of the results is repetitive. Please, just highlight the main information obtained in the results. For example, “At the palatal area, Medit had the highest amount of deviation, followed by the Trios and VPS groups with a P value of 0.0126 and 0.0002, respectively. At the vestibular area, Trios had the lowest amount of deviation, followed by the VPS and Medit groups with a P value of <0.0001. In the overall trueness, Medit had the highest amount of deviation, followed by the VPS and Trios groups with a P value of <0.0001 and 0.0013, respectively”. I would suggest mentioning that the Medit group showed the highest mean values for the three anatomic areas scanned.

-The overall mean for trueness (Table 1) looks different than that which is supposed to be. The overall mean for all groups in the precision (Table 2) shows a correct value. However, the overall mean for the VPS group is wrong. It is supposed to be 42.156. Can the authors clarify the value of 40.930 obtained?

-I suggest not repeating the information of the results already shown in the Tables to the Results section text. The best way to make it easier for the reader is to highlight the highest and lowest means for trueness and precision. Avoid describing the comparable results between groups.

-The authors cannot say that the VPS group showed the highest deviation (which actually means mean) since no statistical significance was detected compared to the Medit scanner.

Discussion

-In the first paragraph, I suggest the authors change the statistical test and reanalyze it to accept or reject the null hypothesis (depending on the results). It is important to consider the two factors: different anatomic areas (ridge, palate, vestibule) and the scanners (VPS, Medit, Trios).

-Check the rules of this journal. Sometimes the authors list the references as numbers and sometimes with the names of the authors. E.g.: Digital impressions must achieve at least the same degree of quality and accuracy as existing conventional procedures to replace the traditional impression and stone cast.31.

-It is important to clarify in the discussion section that a lower reported accuracy (precision) value indicates a higher accuracy of the scanner. This is because accuracy is often expressed as the variation or deviation between repeated measurements.

-The authors say “Trios also showed comparable trueness values to that of the Medit scanner, as in another published article (Michelinakis et al., 2020)”. However, Table 1 does not show comparable trueness mean values between Trios and Medit scanners.

-Again, the authors say that VPS impression exhibited the lowest precision among all tested groups, but it showed statistical similarity with the Medit scanner. Although, these results must be reanalyzed through a different statistical test.

-The authors highlight that the residual ridge and the palatal regions exhibited the highest trueness and precision, respectively, which is not true. Please, revise this information.

-Avoid using abbreviations in the text such as “That’s”.

Additional comments

The manuscript shows an important topic in the dentistry field and shows an adequate methodology. However, major revisions need to be performed such as English grammar, statistical tests, and consequently, update the data obtained.

---

## Round 0.2 · accepted · Accept

Dear authors,

We sincerely appreciate your thorough efforts in addressing all the reviewers' comments and implementing the necessary revisions. After careful evaluation, we are pleased to inform you that your manuscript has been accepted for publication. Congratulations on this achievement, and we look forward to seeing your work contribute to the field.